# Recent Advances in Polymeric Drug Delivery Systems for Peripheral Nerve Regeneration

**DOI:** 10.3390/pharmaceutics15020640

**Published:** 2023-02-14

**Authors:** Marta Bianchini, Silvestro Micera, Eugenio Redolfi Riva

**Affiliations:** 1The BioRobotics Institute, Department of Excellence in Robotics and AI, Scuola Superiore Sant’Anna, 56127 Pisa, Italy; 2Translational Neuroengineering, Centre for Neuroprosthetics and Institute of Bioengineering, School of Engineering, École Polytechnique Fédérale de Lausanne (EPFL), 1000 Lausanne, Switzerland

**Keywords:** peripheral nerve regeneration, polymeric drug delivery systems, microparticles, nanoparticles, nanofibers

## Abstract

When a traumatic event causes complete denervation, muscle functional recovery is highly compromised. A possible solution to this issue is the implantation of a biodegradable polymeric tubular scaffold, providing a biomimetic environment to support the nerve regeneration process. However, in the case of consistent peripheral nerve damage, the regeneration capabilities are poor. Hence, a crucial challenge in this field is the development of biodegradable micro- nanostructured polymeric carriers for controlled and sustained release of molecules to enhance nerve regeneration. The aim of these systems is to favor the cellular processes that support nerve regeneration to increase the functional recovery outcome. Drug delivery systems (DDSs) are interesting solutions in the nerve regeneration framework, due to the possibility of specifically targeting the active principle within the site of interest, maximizing its therapeutical efficacy. The scope of this review is to highlight the recent advances regarding the study of biodegradable polymeric DDS for nerve regeneration and to discuss their potential to enhance regenerative performance in those clinical scenarios characterized by severe nerve damage.

## 1. Introduction

Peripheral nerve injuries (PNIs) cause the disruption of the axonal connection between neural cell bodies and innervated tissue or organ. The peripheral nerve has a fascicular structure, consisting of fibers, connective tissue, and blood vessels. As shown in Figure 1, the axon and myelin sheath form a nerve fiber. The fibers are immersed in loose connective tissue, called endoneurium, and coated with perineurium. The fibers, collected in fascicles, are surrounded by epineurium, a dense connective tissue [1].

The peripheral nervous system can regenerate its fibers. After a complete nerve transection, two stumps are formed, namely the proximal and distal nerve stumps. This type of nerve injury is known as neurotmesis and deeply compromises organs, muscles, and limbs’ functionality. The distal stump undergoes Wallerian degeneration that removes debris. Macrophages migrate and clean up the injured site, and Schwann cells (SCs) promote the formation of a favorable environment for axonal regeneration. SCs can change their state, becoming regenerative phenotypes. This shift causes an increase in the production of neurotrophic factors (NTs) and an interruption in myelin production. To provide a favorable environment for axonal regeneration, SCs align to form the Bands of Büngner [2,3]. SCs are able to survive for eight weeks after injury [4]. Numerous factors limit nerve regeneration such as slow axonal regeneration (1–3 mm/day), the distance between the damaged site and organ target, and the reduced regenerative capability of neurons over time. For these reasons, time and growth factor concentration within the nerve gap are key factors for nerve regeneration [5].

To regenerate an injured nerve, multiple solutions can be adopted, depending on the distance between the proximal and distal stump. When the gap is small, two stumps can be sutured together. When the gap is larger and suturing the stumps would apply tension to them, a nerve graft is used. The gold standard is an autograft, a segment of nerve harvested from the same patient. Autografts provide the same environment as the native tissue, enhancing nerve regeneration and avoiding adverse immune reactions. Nonetheless, autografting has disadvantages, such as nerve/graft dimensions mismatch, and functional damages, since it requires the asportation of healthy nerve portions [6]. However, when the gap is greater than 3 cm, an alternative solution is an allograft. An allograft is a nerve segment harvested from a cadaver that requires a decellularization process. Disadvantages of allograft are a low number of donors and the need for long-term immunosuppressive therapy [6]. For these reasons, technological alternatives such as engineered nerve grafts have been studied in the last decades to cope with the aforementioned limitations of autografts and allografts.

### Nerve Guidance Channel (NGC): Properties and Limitations

A possible alternative to an autograft or allograft is a nerve guidance channel (NGC), a tubular structure that bridges the distal and proximal stumps. Figure 2 shows images of native rat sciatic nerve, implanted NGC, manufactured by freeze-drying [7], implanted NGC manufactured by electrospinning [8] and autograft, 1 year after implantation, [9].

NGC holds the two nerve stumps coaxially at a certain distance between them, providing a biomimetic environment for nerve regeneration and avoids the formation of fibrotic tissue that would prevent axons from regrowing [10]. Regarding this, permeability must allow the passage of nutrients but at the same limit the infiltration of fibroblasts [11]. NGC should be biodegradable, biocompatible, and it should have similar mechanical properties of native tissue. The biodegradability and biocompatibility depend on the materials that were used to manufacture the scaffold. Biocompatible and biodegradable materials allow the avoidance of a second surgery to remove the implant and can reduce the immune response [12]. Mechanical and physiochemical properties of NGC depend on the manufacturing techniques and materials. The type of manufacturing techniques chosen will determine, for example, the porosity, and the presence of nanofibers or micro/nanogrooves [13,14]. Natural or synthetic materials can be used for the manufacture of NGC [10].

The process of nerve regeneration inside a hollow NGC can be divided into five stages. Within hours of the damage, NGC fills with NTs, and, as early as a week later, a fibrin cable forms. Subsequently, SCs migrate to form the Bands of Büngner. After two weeks, axons begin to grow, and three weeks later, maturation and myelination of axons occurs. The main bottleneck of nerve regeneration within an NGC is the gap length. Below a critical gap length, nerve regeneration within NGC occurs spontaneously. However, for longer gaps, regeneration is not efficient, as SCs fail to form the Bands of Büngner, due to a lack of fibrin cable formation [15]. The critical gap length is defined by the distance where successful regeneration may occur 50% of the time after the implant of a traditional nerve graft. The percentage of successful regeneration decreases as the length of the critical gap length increases. The critical gap length in humans and rats is approximately 3 cm and 1 cm respectively [6]. To regenerate nerve damaged over the critical gap, it is important to both accelerate regeneration and maintain a favorable environment for the axonal extension. To enhance nerve regeneration beyond the critical gap length, numerous materials and designs were studied to manufacture an NGC with appropriate characteristics. For example, NGCs with micro/nanofilaments with micro/nanogrooves, with microchannels, or with micro/nano systems for controlled drug deliveries have been studied. A biomimetic NGC with micro/nanoscale topographical cues can promote nerve regeneration as it mimics the structure of native tissue. The presence of aligned micro/nanostructures, such as aligned micro or nanostructures that mimic native tissue structure (microchannels, nanofibers, or microgrooves) promotes SCs migration and axonal growth [16,17,18]. However, to effectively enhance nerve regeneration over a critical gap length, a very promising solution is molecular therapy with controlled DDS. This approach is able to speed-up SCs migration and proliferation onto the fibrin cable that bridges the gap between the two nerve stumps. In this regard, a faster SCs population of the nerve gap is correlated with better regeneration outcomes and functional recovery [19,20,21,22]. Polymeric micro-nanostructures are ideal candidates for this task, as they have been studied for decades to load and release an active principle in a controllable fashion. In the case of peripheral nerve regeneration, the most used active principle is growth factors [15]. Several strategies have been studied in recent years to use DDS for the controlled release of an active principle from NGCs. The following chapters will highlight the most important recent studies on this topic, with the aim of providing a critical overview of this line of research and facilitating its future developments.

## 2. Drug Delivery System for Peripheral Nerve Regeneration

DDSs have been studied for decades to allow the controlled release of an active principle and to maximize its targeting efficacy to overcome the limitations of systemic administration [23,24]. The control of drug concentration allows the avoidance of repeated doses, and consequently, side effects [25]. The choice of biomaterials is very important as their physiochemical properties play a crucial role to control drug release kinetic [26]. Synthetic and natural biomaterials have been employed to fabricate DDSs. Both these classes have advantages and disadvantages. Natural compounds possess similar physiochemical properties to native tissue and their degradation products are known to be safe, on the contrary synthetic materials have better mechanical properties and can be easily processed with various manufacturing techniques, although they are less biocompatible than natural materials [27].

In addition, manufacturing technique is also important, as it can influence system characteristics and drug release profiles [28,29,30]. Furthermore, biomaterials can be tailored to respond to internal (e.g., PH or temperature) and external (e.g., magnetic, electric, or ultrasound stimulus) stimuli to alter their structure and properties, thus controlling drug release [25].

Regarding nerve regeneration, DDSs can be integrated inside the NGC structure to convey drug release in the interstump nerve gap, where the regeneration process takes place. Drug release from NGCs can be divided into two approaches: drug incorporated directly in NGC, or drug incorporated in a micro/nanostructure, loaded in NGC [19,31]. In the first case, a drug can be bonded to biomaterial and the nature of the bond (e.g., covalent, chemical interaction, or physical-chemical interaction) determines the rate at which the drug is released.

The second approach to releasing a molecule from an NGC is much more efficient than the first one, as it allows protection of the drug from degradation due to extracellular fluid exposure and it can be used to tune drug release kinetics. In this case, the release profile depends on the degradation rate of the biomaterial and drug diffusion. By comparing nerve growth factor (NGF) release directly from NGC and NGC-incorporated chitosan microparticles (MPs), it can be observed that MPs significantly reduce the amount of drug released not only in the first three days but also in the overall duration of the experiments. NGF released from MPs, loaded in NGC, and from NGC showed a significant difference [32]. Thus, MPs can not only protect drugs but more importantly can influence drug release kinetics.

By reviewing the literature, three types of DDSs from NGC can be distinguished: MPs, nanoparticles (NPs), and nanofibers. An important parameter for particles and fibers is the diameter since particles’ surface area influences their reactivity with the environment.

### 2.1. Factors Influencing Drug Release from Micro/Nano Systems

Drug release from polymeric micro or nanostructure occurs through three mechanisms: swelling, diffusion, and erosion. Swelling is due to water uptake into a polymeric matrix. When a polymeric matrix hydrates, pore size increases allowing the drug to diffuse out to the external environment. This process can be reduced by using hydrophobic polymers [33]. Diffusion and swelling are strictly correlated because the swelling causes the drug diffusion increase [34,35]. The last process involved in drug release is erosion, which affects the polymeric matrix. Erosion is the degradation of the polymeric matrix due to water action. When water penetrates a polymer matrix, causing its hydrolysis. At first surface erosion occurs, followed by bulk erosion. Diffusion and swelling influence the initial phase of release, and erosion causes the release of the total amount of the drug incorporated within the delivery system [35].

Manufacturing the desired micro-nanostructure allows one to tune its drug release kinetic. Drug release profiles can be divided into two types: single and multiple phases of release. The zero-order release profile is a single-phase process characterized by a constant release of drugs over time [36]. A multi-stage profile may consist of the following phases: burst release, swelling/diffusion phase, delay phase, and degradation phase. A polymeric coating allows the delay of a drug release, obtaining the delay phase. In this case, the drug only diffuses externally after the polymer coating has degraded [36]. The initial phase is characterized by burst release and it refers to the release of a large amount of drug during the initial phase of release (e.g., hours or days) [37]. A burst release is related to water intake that causes swelling and drug diffusion.

In a sustained drug release, the burst release is a problem because it involves the release of a larger amount of the therapeutic agent in a short time. In this case, reducing the burst release is a challenge. Multiple factors influence burst release, and consequently, release kinetics, such as polymer molecular weight, the interaction between drug and polymer, drug loading, and micro or nanostructure size [24,32,36,38]. The molecular weight of the polymer influences the degradation rate and release. To achieve a sustained drug release with a low burst release, a high molecular weight polymer is recommended as it decreases the degradation rate. For instance, Wood and colleagues manufactured poly(lactic-co-glycolic acid) (PLGA) MPs to release glia cell-derived neurotrophic factor (GDNF). Changing the molecular weight of PLGA caused variation in the release duration. Increasing molecular weight the release rate decreased [39]. In the framework of DDS, synthetic materials have been widely used due to their ability to precisely tune drug release kinetic and reduce the burst effect. Another important factor is the interaction between the drug and polymer, which can be weak (electrostatic interaction) or strong (covalent bound). The drug only diffuses through a polymeric matrix when this interaction is broken. Therefore, a strong interaction between the polymer and drug can slow release. This interaction can also be present between drug and cross-linker, used to reticulate the polymer. For instance, chitosan MPs were reticulated with sodium tripolyphosphate and loaded with brain-derived nerve factor (BDNF). To evaluate the release profile of MPs, bovine serum albumin (BSA) was used as a model protein. Comparing the release of two proteins (BDNF and BSA), a difference emerged in the burst release and the total amount released (80.5% for BSA, and 68.8% for BDNF). They assumed that this difference was due to greater ionic interaction between BDNF and cross-linker, as BDNF in the phosphate-buffered saline (PBS) has a positive net charge compared to the negative charge of BSA. Furthermore, BDNF released from MPs increased the percentage of PC12 cells with axonal elongation compared to the samples treated with empty MPs and free BDNF [40]. Drug distribution is another factor that can influence release. If a large percentage of the drug, instead of being loaded inside, is loaded on the surface of the structure, then this can cause a high burst release. Therefore, a possible solution is an MP coating. Figure 3 illustrates the three fundamental aspects that must be considered when implementing a DDS: release profile, polymer-drug interaction, and the release mechanisms involved. Having control of these two aspects allows for tuning the desired release kinetics.

### 2.2. Molecules to Enhance Nerve Regeneration

To enhance nerve regeneration over the critical gap length, multiple molecules can be loaded in micro or nanostructures, such as NTs, anti-inflammatory, immune-suppressant, voltage-gated potassium channel blocker, or erythropoietin (EPO) [41,42].

NTs are proteins able to support nerve cell growth and survival. NTs have a short half-life and are rapidly degraded [43]. NTs include: NGF, BDNF, GDNF, and neurotrophins (NT-3 and NT4/5). NGF acts on sensory neurons and has a great affinity with tyrosine kinases receptor A (TrkA), and GDNF is very effective on motor neurons. BDNF has a great affinity with tyrosine kinases receptor B (TrkB) and supports nerve cells. After an axon transection, neurons and SCs in the distal stump increase the production of NTs. Depending on the entity of nerve injury, this internal source of proteins is not sufficient to support axonal regeneration. In addition, when a nerve defect is over the critical gap length, an external source of NTs is crucial in order to enhance nerve regeneration and further functional recovery [44,45,46].

Multiple studies have been conducted to evaluate the effect of external source of NT on nerve regeneration. In vitro study, NGF was bound to a fibrin gel by heparin. Bhang and colleagues observed that an increase in fibrinogen and thrombin reduced the NGF release rate. The NGF-loaded fibrin gel showed a higher number of neurites per cell and percentage of PC12 cells bearing neurites than the sample treated with NGF added daily [47]. In another study, a silicone tube was filled with a fibrin gel as a delivery system (DS) to release NGF to promote the regeneration of a 13 mm sciatic nerve defect. Three different concentrations of NGF were used to evaluate its effect on nerve regeneration (5, 20 e 50 ng/mL). The two higher concentrations of loaded NGF in the tube with DS showed morphometric results, in terms of the fiber diameter and density, which were not significantly different from isograft [48]. To evaluate the effects of GDNF and NGF on nerve regeneration, a synthetic NGC was used to bridge a 15 mm sciatic nerve defect. In this case, NTs were loaded in the rods of ethylene vinyl acetate copolymer and BSA and subsequently incorporated in the NGC. The results showed that GDNF release promoted nerve regeneration more than NGF [49]. In another study, the effects of GDNF, NGF, and ciliary neurotrophic factor (CNTF) were evaluated on the dorsal root ganglia (DRG) explant. The combination of three NTs with different concentrations showed better neurite outgrowth and neurite length compared to the single factor, showing a dose-response effect [50]. Boyd and Gordon demonstrated that the effect of BDNF on both immediately repaired and chronically axotomized nerves is dose-dependent. They observed that high doses of BDNF did not promote nerve regeneration, whereas low doses did [51]. Santos and colleagues loaded different NTs in PLGA MPs to promote nerve regeneration. PLGA MPs were dispersed in a collagen solution and were used to fill a silicone tube in order to regenerate a 6-mm sciatic nerve defect [52]. In vitro results showed a difference between samples treated with free and encapsulated NTs. They observed that in SC cultures BDNF-loaded MPs had a higher number of neurite lengths, without and with incubation for one week, than collagen without and with free NTs. In DRG cultures, NGF-loaded MPs showed an increase in neurite length only after one week of incubation, compared to other samples. In addition, in vivo results showed the different effects of NTs (BDNF, NGF, GDNF, NT-3, and fibroblast growth factor (FGF)). The number of motor and sensory neurons was evaluated by comparing free and loaded NTs. In the free form of NTs, they observed that BDNF and GDNF promoted motor axon regeneration compared to the control (only collagen), while in the sensory axon regeneration, only NGF showed a difference with respect to the control. The authors observed that MPs promoted axonal regeneration in motor neurons, except FGF, compared to free NTs. Similar results were observed in sensory neurons, except for NGF and BDNF [52].

Tacrolimus (FK506) is an immunosuppressant, and if it is administrated systematically, can cause side effects [53]. In a model of nerve transection and immediate repair, FK506 was released by fibrin gel, as a delivery system [54]. In this study, three different delivery systems were used FK506 loaded in PLGA MPs, particulate FK506, and solubilized FK506. In vivo results showed that FK506-loaded MPs and particulate FK506 enhanced axonal regeneration of motor and sensory neurons, and the number of myelinated axons compared to solubilized FK506 and samples without drug. In addition, biodistribution analyses have demonstrated the efficacy of the delivery system, as no FK506 was detected in vital organs [54]. Yin and colleagues evaluated the synergistic effect of FK506 and NGF administered by subcutaneous injection in the allograft used to bridge a 10-mm sciatic nerve defect [55]. The combination of FK506 and NGF promoted nerve functional recovery. They simulated a hypoxia condition, which occurs following the loss of blood vessels, and observed that FK506 and NGF enhanced the neurite length of PC12 cells in normoxic and hypoxic conditions [55]. In chronic nerve denervation, FK506 does not promote regeneration [42].

Another category of drugs used to promote nerve regeneration is the family of anti-inflammatory drugs. For instance, dexamethasone is able to reduce the foreign body reaction [56]. Dexamethasone is administered to treat several diseases, for example, autoimmune diseases, and to avoid transplant graft rejection. Dexamethasone can act as an anti-inflammatory in two different ways: in one case it acts on the signaling pathways to change cell function in order to avoid the production of inflammatory mediators and proteins. In another case, dexamethasone blocks the signaling pathways to produce inflammatory proteins [57]. After an injury, two types of macrophages migrate to the lesion site: M1 promoter of inflammation and M2 promoter of repair. Macrophages promote nerve regeneration and angiogenesis. In addition, macrophages M2 can promote SC migration and proliferation. Thus, a high ratio of M2/M1 is an important factor to promote nerve regeneration [58]. For instance, thrombomodulin (TM), a transmembrane glycoprotein, promoted macrophage polarization from M1 to M2, inducing a decrease in the expression of M1 markers and an increase in M2 markers. SCs were incubated with M1 cells to generate an inflammatory response and a reduction in the levels of inflammatory markers were observed after TM treatment [59].

The inhibitor of the voltage-gate potassium channel, 4-aminopyridine (4-AP), is another interesting drug. Blocking the potassium channel can increase the duration of action potentials and achieve sustained neurotransmitter release [7].

Table 1 summarizes the molecules to enhance nerve regeneration. As it is possible to notice, NTs resulted to be the most used to enhance nerve regeneration, given their ability to support neural and SCs viability. After molecule encapsulation, the chosen DDS is loaded into the NGC structure to target the release of the active principle in the interstump gap, or directly injected in the damaged nerve, in case of low entity damages.

### 2.3. Polymeric Drug Delivery System for Peripheral Nerve Regeneration

To achieve a controlled and sustained release, DDS can be incorporated into the NGC or directly injected into the damaged nerve. A crucial feature to maximize the efficacy of the molecule is the release kinetic, which can affect nerve regeneration. During the first two weeks after the nerve damage, SCs proliferate and migrate in the fibrin cable formed in the interstump gap that bridges the two nerve stumps. Hence the release profile of the chosen DDS should be maximized within this time range to increase the efficacy of the therapy. In addition, DDSs could allow drugs to be released simultaneously or at separate times. The synergistic effect of biomolecules could improve nerve regeneration compared to the release of a single biomolecule. DDSs can be trapped within the structure of an NGC or injected directly into a damaged nerve.

#### 2.3.1. Injection of Drug Delivery Systems within the Damaged Nerve

DDS could be directly delivered within a crushed nerve in the case that the entity of the damage does not require an NGC to be bridged. For instance, Zhang and colleagues injected microspheres (MSs) into the sutured nerve. They manufactured PLGA MPs loaded with EPO and BSA-PLGA MSs loaded with NGF. The presence of BSA in NGF-loaded MPs slowed release in the first three days, as shown in Figure 4c, to reduce the apoptotic cell death caused by intraperitoneally administrated NGF immediately after damage. The synergistic effect of EPO and NGF showed better results in terms of histology and electrophysiology than empty MSs and MSs loaded with a single drug [60]. Injection of DDS into drug release can be used to regenerate a crushed nerve. To regenerate a crushed nerve, Manto and colleagues injected into the crushed sciatic nerve a thermosensitive PLGA-b- polyethylene glycol (PEG) triblock copolymer, that in an aqueous solution forms micelles, containing 4-AP [61]. The sol-gel transition temperature varied with the concentration of 4-AP but gelling occurred in the range of the body temperature. The release duration was about 28 days. Copolymer with 4-AP showed a higher amount of neurofilament-H and myelin protein zero markers than copolymer without drug and injection of drug systemic administration. Similar results were observed in the analyses of functional sciatic index [61].

DDS for peripheral nerve regeneration could also be used to reduce the side effects of systemic drug administration. For instance, FK506, an immunosuppressant usually used during transplantation, can be used to avoid immune rejection after allograft implantation, but it can cause severe hepatic toxicity or mutagenic effects if administrated systemically. For this reason, the group of Zuo and colleagues encapsulated it in PLGA MPs to reduce immune rejection following allograft implantation [62]. The results obtained from local (injected onto the allograft surface) and systemic (intraperitoneal injection) administration of FK506-loaded MPs, were compared with each other and with those obtained from isograft implantation, as a positive control. The results showed that FK506 released locally and systematically enhance the number of motor and sensor neurons, in the midgraft, myelinated axons in the midgraft, and the distal nerve compared to allograft untreated, and were no different from results obtained with the isograft. In addition, the allograft treated with local and systemic FK506 showed a reduction of the proinflammatory cytokine. Moreover, FK506 loading in MPs allows a drastic reduction of the administrated drug with respect to systemic injections (about 50 times less) [62]. The same context evaluated the effect of GDNF release from PLGA MPs in allograft implantation [63]. The molecular weight and viscosity of MPs polymer can influence release. An increase in poly PLGA molecular weight can extend the duration of GDNF release up to 28 days, as shown in Figure 4e. In this case, GDNF-loaded MPs, with a duration of two weeks and four weeks, were incorporated in a fibrin gel that was injected in the proximal and distal stump, after acellular nerve allograft implantation, as shown in Figure 4d. Allograft was used to bridge a 5 mm sciatic nerve defect. Better results were obtained from samples treated with the two week formulation, with respect to the samples without DDS, which were used to perform histomorphometric analysis. Then, DDSs enhance nerve regeneration with respect to samples without DDS [63].

#### 2.3.2. DDS Incorporated within NGC Structure

In the event of nerve damage that caused a consistent loss of tissue, an NGC is implanted to hold the two-nerve stump coaxially and provide a favorable environment to promote regeneration. In this scenario, a DDS can be loaded inside the wall of the conduit or within the interstump gap, in order to favor the release of the bioactive molecule in the intraluminal space of the NGC. Several designs of polymeric DDS have been studied nowadays and can be classified into three subgroups: MPs, NPs, and nanofibers.

##### Microparticles

MPs loaded with drugs and incorporated into NGCs represent one of the most widely studied DDS for nerve regeneration purposes. MPs dimensions, type of polymer, and the presence of single or multiple polymer shells, as well as manufacturing technique can influence drug release and consequently impact nerve regeneration performance. Regardless of the overall release profile, the burst release should be reduced to avoid side effects due to excessive molecule release concentration and drug dispersion in the very initial stage of regeneration that could impede its release over longer periods. When MPs are incorporated within an NGC, polymeric layers of NGC represent an additional barrier that must be crossed by the drug to reach the target site.

Polymer coating for MPs surface has been a widely studied strategy to reduce burst release. Fadia and colleagues fabricated core-shell MPs embedded in a polycaprolactone (PCL) NGC to release GDNF [9]. NGC was used to bridge a 5 cm median nerve defect in nonhuman primates. MPs consisted of a poly(L-lactic acid) (PLLA) shell and a PLGA core, containing GDNF. After one year, NGC with GNDF-loaded MPs showed better results in terms of quantification of SCs, neurofilament, and G-ratio than the NGC without GDNF. Similar results were observed when evaluating the compound nerve action potential (CNAP) and the compound muscle action potential (CMAP) [9].

Moreover, a coating can be manufactured to buffer the pH. In these cases, the polymer coating can also have the purpose of buffering the pH [64]. In the preparation of MPs by ionotropic gelation, the amount of crosslinker can influence the properties of microparticles and release. By increasing the amount of crosslinker, SI and size are reduced [65]. An increased crosslinker concentration forms a denser microparticle structure that not only slows down drug diffusion but also causes a reduction of EE due to reduced free space to encapsulate the drug [66,67].

Zeng and colleagues manufactured a collagen-chitosan NGC loaded with NGF incorporated in chitosan MPs [32]. The release profile of NGF-loaded MPs incorporated in the NGC (Figure 5a–d) showed a reduced burst release compared to unloaded MPs, as shown in Figure 5e. NGC was used to bridge a 15 mm sciatic nerve defect. After 16 weeks, the amplitude of CMAP and the nerve conduction velocity (NCV) in the NGC with NGF-loaded MPs were higher than NGC with free NGF and without NGF. In addition, NGF released by MPs, incorporated into NGC, increased the diameter of myelinated axons and the total area of regenerated axons compared to NGC with free NGF and without [32].

In another study, an NGC was loaded with PLGA MPs containing NGF and GDNF respectively to regenerate a 15 mm sciatic nerve defect. NGF and GDNF effects on cells showed a time-dependent mode. Lackigton and colleagues observed that a reduced initial burst release promotes axonal growth and SC migration [68]. This increase was observed to a greater extent in the presence of both NTs. MPs loaded in the NGC showed that the burst release and rate release are reduced, as shown in Figure 5f,g. In vitro results showed that NGF and GDNF encapsulated and loaded into the NGC promoted more neurite growth, SC growth, and proliferation (indicated by the metabolic activity) than non-encapsulated NTs. Promising results were also observed in animal tests in terms of functional recovery and muscle mass. In fact, NGC with NGF and GDNF encapsulated had better results than NGC without NTs but similar to autograft [68].

Table 2 summarises some previously mentioned studies of MPs for peripheral nerve regeneration.

##### Nanoparticles

Nanoparticles (NPs) are another interesting example of DDS used for nerve regeneration purposes. NPs size is important to prevent a fast clearance but also to favor biodistribution.

As with microparticles, a method to reduce the burst release and extend the release duration is to create a coating around the particle. An important cellular process is chemotaxis, which drives the migration of cells from a space with a lower concentration of NTs to a space with a higher concentration. Therefore, a possible solution to favor cell migration and promote nerve regeneration is to create a drug concentration gradient [2]. A drug concentration gradient within NGC would allow SCs to migrate and populate the distal stump faster. There are two popular methods for creating a drug concentration gradient: one is to bind the drug directly to the structure of NGC, while in the second micro or nanoparticles were dispersed with a different concentration in the polymeric solution used to make the NGC [69,70]. For instance, Chang and colleagues [71] fabricated a multichannel NGC with aligned nanofibers and a gradient of NT concentration to bridge a 15 mm sciatic nerve defect. To slow the release of NTs, NGF was incorporated into NGC, and BDNF was encapsulated in gelatin nanoparticles, and loaded into NGC. This occurrence allowed for a slower release of BDNF with respect to NGF, which has to diffuse out of the NPs structure [71]. This difference in the release of the two factors is shown in Figure 6a, where it is possible to notice slower BDNF release with respect to NGF at different time points. Hence selective NPs encapsulation is a promising strategy not only to slower drug release, but also to selectively tune the release of multiple growth factors from a single scaffold.

An interesting subset of NPs is represented by nanotubes, which have been widely presented as DDS [72]. For instance, Manoukian and colleagues [7,73] loaded halloysite nanotubes (HNTs) with 4-AP that were encapsulated in a porous NGC to enhance nerve regeneration. NGC was fabricated to bridge a 15 mm sciatic nerve defect. By choosing chitosan as an NGC material, the electrostatic interaction between positive groups of chitosan and negative groups of HNTs can avoid the diffusion of nanostructures outside of the conduit. The results showed that drug loading within HNTs can prolong the release with respect to drug loading directly inside the NGC structure. In this regard, drug release from NHTs without conduit lasted 6 h but when NHTs were incorporated into NGC, the duration was approximately 168 h (in the first study) [7] and, in the second study, eight weeks [73]. They prolonged the study of 4-Ap release. In vivo results showed that NGC with the drug can enhance functional recovery and nerve regeneration compared to NGC without the drug [73]. The difference in the burst release between samples is shown in Figure 6b.

An innovative strategy to enhance nerve regeneration over the critical gap length consists of incorporating magnetic nanoparticles into the scaffold. By applying a magnetic field, the position of magnetic nanoparticles can be controlled to create topographical cues within the NGC that have been demonstrated to have a positive effect on nerve regeneration [74,75]. In addition, magnetic nanoparticles (MNPS) and magnetic fields can be used to generate tensile forces that are able to stimulate axonal growth [75]. In this case, magnetic nanoparticles were coated with PEG coupling, with EDC-NHS allowing them to bind with NGF, as shown in Figure 6c [76]. MP-bound NGF were dispersed in a collagen gel, and subsequently, crosslinked with genipin. This gel was used to fill the lumen of the NeuraGen^®^ (Integra LifeSciences, Princeton, USA) conduit to regenerate an 8 mm sciatic nerve defect. MNPs were used to create aligned collagen fibers and control NGF distribution by applying an external magnetic field. In vitro results showed that the MNPs with and without NGF, internalized by the cells, increased the PC12 elongation ratio compared to the sample with free NGF. In addition, MPNs with NGF enhanced the average number of axons and showed better values of sciatic index function than the conduit without NGF, empty, and with free NGF [76]. Table 3 summarizes some previously mentioned studies.

##### Nanofibers

Nanofibers are another widely used example of DDS for nerve regeneration purposes and can be manufactured by three techniques: self-assembling, phase separation, and electrospinning. Electrospinning is a straightforward manufacturing technique that allows random or aligned fibers to be made with different diameters. Fiber characteristics depend on manufacturing parameters and the type of polymer. Electrospinning is capable of creating three different fibers: uniaxial, coaxial, and triaxial [77].

As for other DDS, drugs can be loaded in nanofibers during the process or post-electrospinning. For instance, a PCL NGC was fabricated with aligned fibers in the inner layer and random fibers in the outer layer. After electrospinning, the surface of NGC was modified by plasma treatment to create a gradient distribution of amino groups (NH_2_) to which heparin was bound, acting as a bridge between the NH_2_ group and NGF, as shown in Figure 7a [78]. In vitro results showed that aligned fibers and concentration gradient enable unidirectional axonal growth, as shown in Figure 7b. This NGC was used to bridge a 15 mm nerve defect. The in vivo results showed the functional recovery of NGC with aligned fibers, and a concentration gradient higher than NGC without a concentration gradient and NGC with uniform distribution of NGF [78]. Thus, topographical, and biochemical cues play a critical role in nerve regeneration, enabling greater axonal growth with preferential direction. With the same method, the simultaneous release of different drugs can be achieved. PCL NGC was fabricated by electrospinning to release NGF and BDNF simultaneously. In this study, growth factors were immobilized to the NH_2_ groups by heparin to slowing molecule release. The presence of heparin reinforced the interaction, resulting in a release of 21 days compared to 7 days. The release of individual growth factors resulted in a greater neurite extension and SC migration than simultaneous release. Probably, this difference is due to the higher affinity of receptors for NGF compared to BDNF in a chicken embryo DRG [79].

Drug release by breaking the chemical bond between the molecule and the polymeric carrier is a very simple method to create a concentration gradient and release multiple drugs simultaneously. At the same time, the drug not being incorporated within the nanostructure can be degraded quickly and its release can show a high burst release. Drugs, incorporated in the same NGC, can be released differently, and with different timing. A preliminary study showed the effects of the simultaneous and individual release of NGF and CNTF from silk nanofibers [80]. Dinis and colleagues showed that the incorporation of NTs into nanofibers reduced burst release. In fact, they observed no factor-loaded nanofibers release until five days. An increase in the glial cell/neurons ratio and neurite extension was observed in the sample treated with NGF and CNFT-loaded nanofibers in comparison to nanofibers without NTs [80].

In another study, NGF and vascular endothelial growth factor (VEGF) were released at different times. NGF was loaded into poly PLLA nanofibers, while VEGF was immobilized on the surface of NGC. VEGF release has a high burst release and a shorter duration than the NGF release. This different timing is due to the incorporation of NGF within nanofiber. This NGC was used to repair a 10 mm sciatic nerve defect. The dual release of NGF and VEGF improved nerve regeneration and revascularization better than NGC without growth factors and NGC with a single growth factor [81]. Another way to achieve simultaneous release is to incorporate drugs into nanofibers. Furthermore, to tune the drug release profile, nanofibers can be made from different polymers to control degradation. For instance, NGF and GDNF were released from poly (D, L- lactic acid) (PDLLA) and PLGA nanofibers, respectively. Nanofibers had an aqueous core with a drug and a polymeric shell. The results showed that NGF release was slower than GDNF one, probably due to the higher hydrophobicity of PDLLA [82,83]. To evaluate the effects of released growth factors on cells, a scaffold with different ratios of GDNF/PLGA fibers to NGF/PDLLA was prepared. In vitro results showed that cell differentiation and neurite outgrowth increased with the release of both growth factors and with more NGF/PDLLA fibers than GDNF/PLGA fibers in the scaffold [83]. To reduce the burst release and slow down drug release, a multi-layered NGC can be a solution. In exploiting this kind of structure, drugs can be released simultaneously or sequentially, playing with the degradation rate of materials [84].

In a different study, a PLGA NGC was manufactured with FK506 to bridge a 15 mm sciatic nerve defect [85]. FK506 was added to the polymeric solution to create nanofibers. By comparing FK506-loaded NGC with autologous implant and empty NGC, functional recovery of FK506-loaded NGC, in terms of the sciatic functional index and compound muscle action potential, was better than empty NGC and was similar to autologous implant. These results were also observed in terms of the SC marker [85].

Chen and colleagues [86] manufactured an NGC to bridge a 15 mm sciatic nerve defect. NGC was manufactured with aligned nanofibers, magnetic nanoparticles (Fe_3_O_4_), and drugs, as shown in Figure 7c. In this case, the loaded drug is melatonin (MLT), an anti-inflammatory drug with an antioxidative effect. Magnetic nanoparticles were incorporated to increase mechanical properties. Two types of NGC were tested: one with a single-layered scaffold, in which nanoparticles and drugs were loaded in PCL nanofibers, and another one with a multi-layered scaffold. The multi-layered scaffold consisted of three layers: an outer PCL layer, a middle layer with magnetic nanoparticles, and an inner layer with MLT. The results showed that a single-layered scaffold released melatonin slower than a multi-layered scaffold. This is probably due to the presence of MLT only in the inner layer, which degrades and causes MLT diffusion. In the first case, the single-layered scaffold, the structure of the NGC slows down the diffusion of MLT, which must pass through a greater thickness than the multi-layered scaffold [86]. It is interesting to observe the contribution in the nerve regeneration of single components, such as magnetic nanoparticles and drugs. Iron oxide nanoparticles stimulate neurite extension because, in the presence of NGF, they are able to trigger the activation of mitogen-activated protein kinase (MAPK) [87]. Table 4 summarises some of the previously mentioned studies.

As mentioned before, the most widely used DDSs are divided into MPs, NPs, and nanofibers and their efficacy depend on multiple factors. In our view, nanofibers have a dual function compared to other types of DDSs. They not only allow the drug to be released along the entire length of the NGC but also create a biomimetic structure that mimics the native tissue structure and guides axon regrowth. Thus, this dual effect could further promote nerve regeneration over the critical gap length.

## 3. Challenges and Future Perspectives

As a consequence of a nerve injury, the autograft is currently the best solution to promote nerve regeneration over the critical gap length. However, as previously mentioned in the Introduction, engineered NGCs represent a very promising alternative to overcome the limitations of autologous transplant. Another important aspect is that the regenerative capacity of SCs decreases over time. To prevent muscle atrophy and loss of motor and sensory function, a faster and more efficient regeneration process is required. For instance, damage to the brachial plexus, a nerve plexus that innervates the upper limb, can cause severe motor and sensory loss over the entire upper limb, with detrimental consequences for the patients [88]. To regenerate such a long nerve defect, strategies that support the cellular processes involved in nerve regeneration are deeply required. The studies reviewed in this work demonstrated that NTs therapy with DDS, incorporated within the NGC structure, represents a promising strategy to enhance nerve regeneration over a critical gap size. Depending on the drug mechanism of action, the release profile should have a different duration to maximize its efficacy. For instance, NT delivery should have a slow release kinetic, that should last for at least two weeks or a month to promote SCs migration and proliferation within the interstump gap. This process is crucial to faster neurite regrowth and further improve functional recovery. Currently, multiple solutions to tune the release profile have been presented. For instance, shell formation and DDS structural modifications are promising strategies to slow down the release and reduce the burst effect, in order to extend the duration of the molecule release over the required time to support the regenerative process.

In this review, the most relevant studies on the release of biomolecules capable of promoting nerve regeneration over the critical gap length have been reported. As mentioned before, the drug can be released from DDS loaded into the NGC or directly from the NGC. The main advantages of DDSs from NGC are drug protection from the external environment, reduced burst release, and greater control over the release kinetics. In particular, to have more control over the release profile, attention must be paid to the structure of DDS. For instance, core-shell structures allow not only to slow down the release profile and to reduce the burst [9,64], but also allow for releasing two different drugs at different times [82,83]. To obtain different release profiles, another very interesting strategy is to load one drug in the structure’s core and the other on the structure’s surface [81]. Three aspects are crucial in the DDSs release profile, release mechanisms, and polymer-drug interaction. Good knowledge of the chemical structure of the drug and the target site, where the drug is to be released, which enables the selection of the appropriate polymer and technique to manufacture the correct DDS formulation. In the context of nerve regeneration, there are multiple challenges. The main task is to fabricate an NGC capable of achieving results similar to those of autograft. To achieve this, a combination of strategies can be employed to promote nerve regeneration. As for the NGC, research into materials and manufacturing techniques is crucial to obtain the optimal DDS.

Among all the reviewed DDS formulations, nanofibers are particularly promising since they allow the creation of a topographical cue similar to the structure of the native tissue. The combination of the topographical cue with the concentration gradient appears to be a very promising strategy to accelerate and promote nerve regeneration over the critical gap length [16,69].

Another important aspect is the chosen type of drug. Many studies have focused on NTs’ effects on nerve regeneration. Immunosuppressive drugs can improve allograft implantation to reduce immune response due to the patient’s rejection [7]. Moreover, anti-inflammatory drugs can be used when constituent materials of the NGC are not biodegradable, such as silicone. In this case, the drug would reduce the inflammatory process resulting from the permanence of the NGC. To impede the inflammatory process, the polarization from M1 to M2 must be promoted [58]. Above the discussed ones, NTs represent the most promising molecules to enhance nerve regeneration, giving their ability to sustain neural cell viability and growth. The reported studies show that the presence of drugs released in a controlled and sustained manner can promote nerve regeneration. Furthermore, it has been observed that the NGC with DDSs allows for greater results in terms of nerve regeneration than NGC without DDSs.

All the reviewed studies demonstrated the strong potential of DDS to accelerate and improve nerve regeneration. However, more efforts are required to have greater control over drug release kinetics. Therefore, an interesting challenge is to develop DDS able to trigger drug release within chosen intervals of time. As an example, an interesting approach could be to combine an external stimulus with DDS to further improve regeneration outcomes. Magaz and colleagues manufactured a silk membrane loaded with graphene oxide (GO) and NGF [89]. NGF loading was carried out by application of potential resulting in active/electrochemical loading. The application of electrical stimulation increased the amount of NGF released by 5 to 8 times compared to the release without stimulation by diffusion [89]. In this view, it could be interesting to develop a stimuli-responsive DDS to better tune the amount of released growth factor, by triggering its release in particular time intervals. In addition, Zhang J and colleagues [90] fabricated core-shell nanofibers with a shell of poly(L-lactic acid-co-ε-caprolactone) (P(LLA-CL))/silk fibroin and polyaniline with a core of NGF. NGF release was evaluated with and without electrical stimulation. The results showed that electrical stimulation increased the amount of NGF released [90]. In another study, Huang and colleagues [91] manufactured an NGC with PLGA microcapsules loaded with magnetic nanoparticles and NGF (Figure 8). PLGA microcapsules were incorporated in a silk/gelatin solution. The application of an external magnetic field had a dual purpose: to create aligned topographical cues, and to trigger NGF release by means of a temperature rise. By applying a high-frequency magnetic field (HFMF), the Neel relaxation process caused a temperature increase that triggered drug release by expanding the PLGA shell, thus, favoring drug diffusion. In addition, the authors studied how magnetic field parameters, gelatin viscosity, and temperature increase influenced particle distribution. They observed that by increasing the temperature above 20 °C MPs were able to move under magnetic guidance within gelatin solution. As the temperature dropped under 20 °C, the microcapsules were immobilized. The viscosity of the gelatin solution influenced the MPs’ ability to move [91]. In vitro results showed that HFMF stimulation allowed the enhancement of neurite length for magnetic nanoparticles-loaded samples with respect to samples treated with NGF-free and control. Moreover, in vivo results showed that a combination of magnetic microcapsules and NGF release improved nerve regeneration performances and functional recovery with respect to samples without HFMF, the sample treated with non-patterned NGF-loaded microcapsules, and control [91].

The application of an external stimulus (i.e., electrical current or magnetic field) was able to modify the physiochemical properties of the polymer carrier, thus enhancing drug release during the time interval of the application of the stimulus. This would allow boosting of the drug release in the event of an unsatisfactory nerve regeneration outcome revealed by standard electrophysiological follow-up. Hence, one could choose to activate drug release from a stimuli-responsive micro- or nanostructure loaded inside the NGC to enhance nerve regeneration in a chosen time interval. This occurrence would enable tuning the amount of biomolecules released to promote nerve regeneration and improve functional recovery.

All these strategies have been reported to be advantageous for nerve regeneration, and future studies will be needed to assess the possibility of better controlled drug release profiles in animal experiments of nerve regeneration over the critical gap length. Although some examples of nerve conduits have received market approval [92], there is a strong need to push forward the investigation on drug delivery by NGCs, giving the aforementioned problem to improve nerve regeneration over the critical gap length. This last occurrence is crucial to explore the effectiveness of DDS in effectively tackling the bottleneck of nerve regeneration, thus pushing forward the clinical translation of DDS-loaded nerve conduits.

## 4. Conclusions

This review highlighted how DDS could effectively improve nerve regeneration performances over the critical gap length. By loading a chosen therapeutic molecule within a polymeric micro-nanocarrier, it is possible to tune its release kinetics in order to aid the cellular process involved in neurite regrowth and consequent reinnervation. NTs are the most promising candidate to accelerate nerve regeneration, due to their ability to support SCs viability. Above the reviewed DDS, nanofibers appeared to be the most promising polymeric carrier, since they could both encapsulate and deliver a drug and provide a topographical cue to promote cell adhesion and axon reinnervation. VEGF has also demonstrated some positive effects to support the revascularization process during nerve regeneration.

In conclusion, this review supports the benefits of employing biodegradable polymeric DDS to support nerve regeneration, especially in case of a consistent lesion over the critical gap length, and would encourage the investigation of this aspect to pave the way to the clinical translation of new nerve regeneration technologies.

## Figures and Tables

**Figure 1 pharmaceutics-15-00640-f001:**
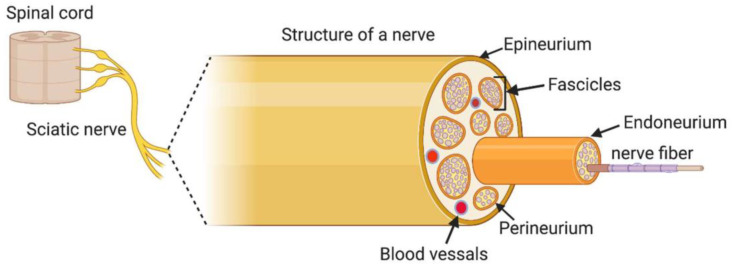
Structure of a peripheral nerve. Reproduced with permission from [1], Copyright © 2020 Published by MDPI.

**Figure 2 pharmaceutics-15-00640-f002:**
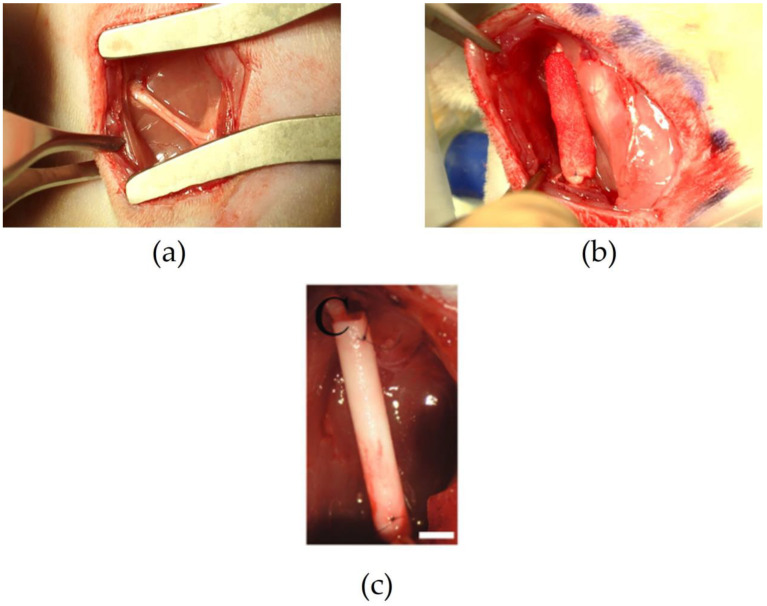
Images of native nerve, implanted NGC, and autograft. (**a**) Native rat sciatic nerve. (**b**) Implanted NGC to regenerate 15 mm sciatic nerve defect.Adapted from/Reproduced with permission from [7]. Copyright Elsevier, 2019. (**c**) Implanted NGC manufactured by electrospinning. Adapted from/Reproduced with permission from [8]. Copyright John Wiley & Sons, 1999.

**Figure 3 pharmaceutics-15-00640-f003:**
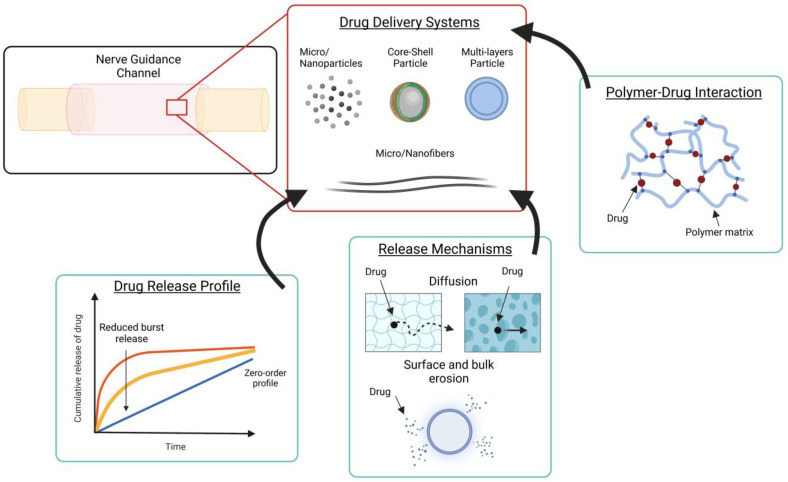
Schematic illustration showing factors that accelerate and promote nerve regeneration by DDS employment. DDS can be incorporated into the NGC to release drugs (i.e., neurotrophic factors, anti-inflammatory drugs, or immunosuppressant drugs). To control drug releases from DDS incorporated within NGC structure, three factors have to be considered (enclosed in green boxes and linked with solid arrows to DDS box in red): drug release profile, release mechanism, and polymer-drug interaction. Created with BioRender.com (3 January 2023).

**Figure 4 pharmaceutics-15-00640-f004:**
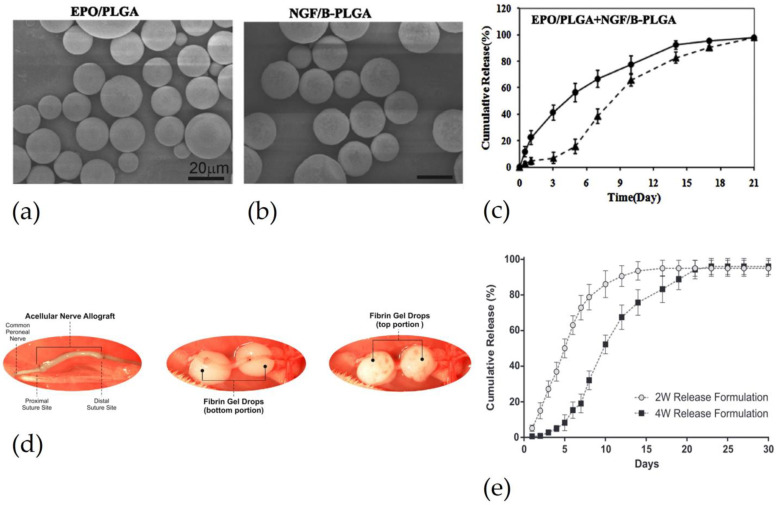
DDS directly injected in the damaged nerve: (**a**) PLGA MSs loaded with erythropoietin. (**b**) BSA-incorporated PLGA MSs loaded NGF (**c**) EPO release from PLGA MSs and NGF release from BSA-PLGA MSs. Reproduced and adapted with permission from [60], Copyright © 2017 Published by Elsevier B.V. (**d**) Procedure to inject fibrin gel, as a drug delivery system. The first image shows a cellular nerve allograft used to bridge a 5 mm nerve gap. The second and third images show the injection of fibrin gel in the top and bottom portions, respectively, of the proximal and distal stumps. (**e**) Comparison of GDNF release between PLGA MPs with different molecular weights: light dot indicates PLGA microparticles with lower molecular weight and dark square PLGA microparticles with high molecular weight. Reproduced and adapted with permission from [63], Copyright © 2015 Acta Materialia Inc. Published by Elsevier.

**Figure 5 pharmaceutics-15-00640-f005:**
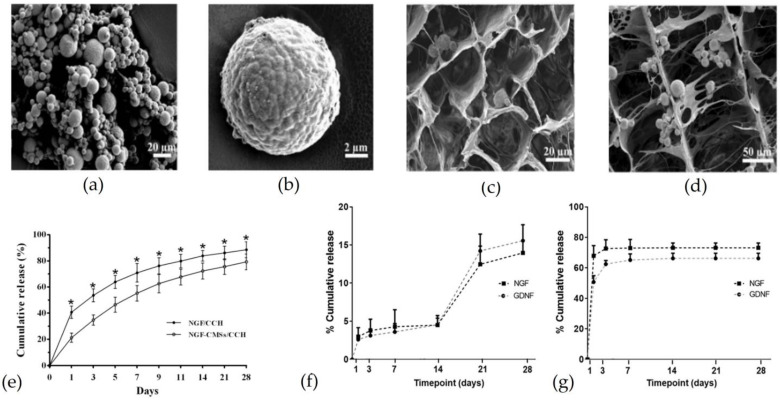
Microparticles as drug delivery systems loaded in nerve guidance channels: (**a**,**b**) Scanning electron microscopy images of chitosan MPs loaded with NGF. (**c**) SEM images of a transverse section of NGC loaded with NGF-loaded chitosan MPs. (**d**) SEM images of a longitudinal section of NGC loaded with NGF-loaded chitosan MPs. (**e**) Comparison of NGF release from MPs embedded in the NGC (empty dot) and release of NGF loaded directly in the NGC (* = *p* < 0.05). Reproduced and adapted with permission from [32]. Copyright © Public Library of Science (**f**) NGF and GDNF release from PLGA microparticles loaded in a conduit, (**g**) NGF and GDNF release from a conduit without PLGA microparticles. Reproduced and adapted with permission from [68], Copyright © 2019 Elsevier B.V.

**Figure 6 pharmaceutics-15-00640-f006:**
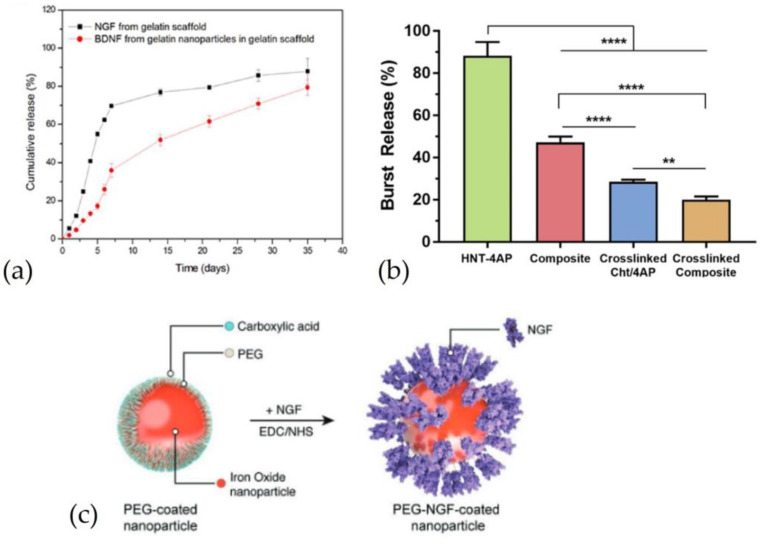
Nanoparticles as drug delivery systems loaded in nerve guidance channels: (**a**) Comparison of release between NGF, incorporated in the scaffold without nanoparticles, and BDNF, incorporated in gelatin nanoparticles loaded in gelatin scaffold. Reproduced and adapted with permission from [71], Copyright© 2017 ACS Publications. (**b**) Comparison of 4-aminopyridine (4-AP) release from halloysite nanotubes (HNT-4AP), HNTs incorporated in the chitosan conduit without crosslinked (composite), chitosan conduit crosslinked (crosslinked Cht/4AP) and HNTs incorporated in the chitosan conduit crosslinked (crosslinked composite) (** = *p* < 0.01; **** = *p* < 0.0001). Reproduced and adapted with permission from [7], Copyright© 2019 Elsevier B.V. (**c**) Representation of magnetic nanoparticles coated with PEG and linked to NGF. Reproduced and adapted with permission from [76], Copyright © 1999–2023 John Wiley & Sons, Inc.

**Figure 7 pharmaceutics-15-00640-f007:**
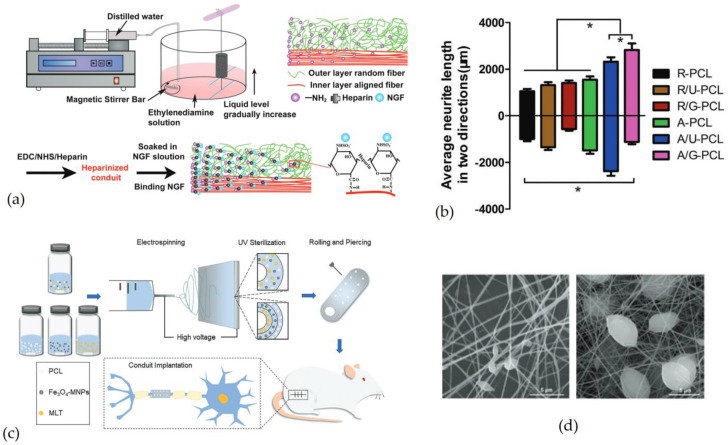
Nanofibers as drug delivery systems loaded in nerve guidance channels: (**a**) Representation of the NGC manufacturing process to create an NGF concentration gradient. Formation of NH_2_ groups, in a gradient configuration, on PCL fibers in order to bind NGF by heparin. (**b**) Representation of average neurite length in two directions: random PCL fibers conduit (R-PCL), random PCL fibers conduit with uniform NGF (R/U-PCL), random PCL fibers conduit with gradient NGF (R/G-PCL), aligned PCL fibers conduit (A-PCL), aligned PCL fibers conduit with uniform NGF (A/U-PCL) and aligned PCL fibers conduit with gradient NGF (A/G-PCL). (* = *p* < 0.01) Reproduced and adapted with permission from [78], Copyright © 1999–2023 John Wiley & Sons, Inc. (**c**) Representation of the single-layered and multi-layered NGC manufacturing process by electrospinning. NGC was loaded with magnetic nanoparticles (Fe_3_O_4_ MNPs) and MLT, an anti-inflammatory drug. (**d**) Scanning electron microscopy images of polycaprolactone nanofibers and composite nanofibers, respectively. Reproduced and adapted with permission from [86], Copyright © 1999–2023 John Wiley & Sons, Inc.

**Figure 8 pharmaceutics-15-00640-f008:**
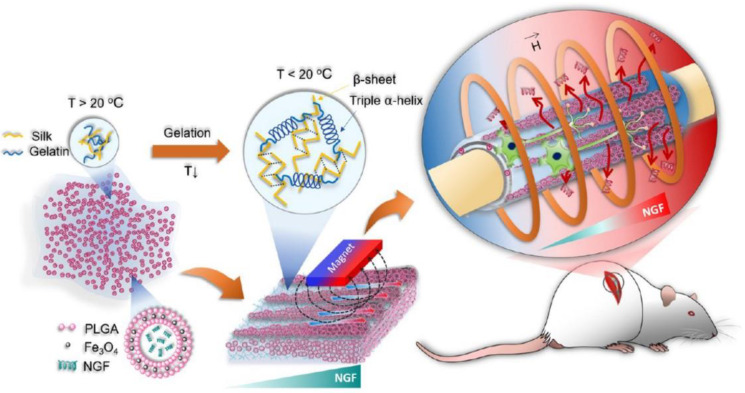
Schematic representation of the manufacturing process to produce an NGC with PLGA microparticles loaded with NGF and magnetic nanoparticles (Fe_3_O_4_). The application of HFMF was able to trigger drug release over a chosen interval of time and indicated by the red arrows. Reproduced with permission from [91], Copyright © 2022 ACS Publications.

**Table 1 pharmaceutics-15-00640-t001:** A schematic list of drugs that promote nerve regeneration.

Drug	Effects	References
*NGF*	Promote survival and growth of sensory neurons	[2,43,46]
*BDNF*	Promote survival and growth of sensory and motor neurons	[2,43,46]
*GDNF*	Promote survival of motor neurons	[2,43,46]
*EPO*	Neuroprotective	[42]
*4-AP*	Inhibitors of potassium channel	[7]
*FK506*	Immunosuppressant and neuroprotective	[53]

**Table 2 pharmaceutics-15-00640-t002:** Schematic list of microparticle studies to enhance nerve regeneration.

Drug	Material	Total Drug Released	Duration Release	Results	Ref.
GDNF	PLGA-Chitosan (2%) (core-shell MPs)	76.8 ± 2.1%	84 days	Chitosan shell reduces burst release; After 28 days, samples with core-shell MPs showed a higher percentage of PC12 cells with neurite extension than a sample with MPs without a shell;	[64]
NGF	Chitosan crosslinked with tripolyphosphate (TPP) loaded in Chitosan-PCL NGC	Up 60%	6–7 weeks	The presence of a Chitosan-PCL layer reduces burst release and slows the release rate of NGF; Increasing the amount of PCL in NGC and TPP, the release rate can be reduced;	[65]
NGF	PLGA MPs loaded with NGF inside Chitosan MPs, loaded with NGF free	67.7 ± 1.2% (1% TPP) 48.5 ± 0.7% (5% TPP) 45.7 ± 0.8% (10% TPP)	After 49 days	Increasing percentage of TPP, as a crosslinker of chitosan, burst release, and release rate can be reduced; MPs with TPP 5% stimulate neurite extension of PCL12 cells more than the negative control (without NGF); The average muscle fiber area of PCL NGC with MPs was similar to autograft. PCL NGC with MPs accelerates nerve regeneration than to PCL NGC and PCL NGC with NPs;	[67]
NGF and GDNF	PLGA	13.95% (NGF) 15.56% (GDNF)	28 days	Without PLGA MPs the release showed a large burst release, and the release duration was 7 days; NGF and GDNF released from MPs incorporated in NGC improve neurite outgrowth; NGC with MPs showed results in terms of functional recovery higher than NGC with empty MPs and similar to autograft. The same results also evaluate muscle weight;	[68]

**Table 3 pharmaceutics-15-00640-t003:** Schematic list of nanoparticle studies to enhance nerve regeneration.

Drug	Material	Total Drug Released	Duration Release	Results	Ref.
NGF BDNF	NGF free and BDNF loaded in gelatin NPs	NGF: up 80% BDNF: up 60%	35 days	Gradient distribution and combination of NGF and BDNF increased cell density and neurite length; Neurotrophic gradient increased the expression of MBP, a marker of myelin; NGC with neurotrophic gradient improved nerve regeneration in terms of histology and electrophysiology;	[71]
4-AP	Hallosyte nanotubes (HNTs)	About 80% 8 (weeks)	8 weeks	Chitosan NGC crosslinked and loaded with NHT-4AP reduced burst released; 4AP released increased the production of NGF and BDNF in human SCs; NGC with drug showed similar results to autograft and better results than NGC without drug;	[73]

**Table 4 pharmaceutics-15-00640-t004:** Schematic list of e micro/nanofiber studies to enhance nerve regeneration.

Drug	Material	Total Drug Released	Duration Release	Results	Ref.
NGF and BDNF	PCL nanofibers with amine groups and heparin bind NGF and BDNF	NGF: max 1 ng/mL BDNF: max 1 ng/mL	21 days	The bond between heparin and growth factor allows for a slowed release rate. NGF with a concentration of 1 ng/mL promoted neurite extension ad SCs migration; The combination of NGF and BDNF did not favor neurite growth and SCs migration (also by increasing the concentration);	[79]
NGF and VEGF	NGF loaded in the core of PLLA nanofibers while VEGF absorbed in the surface	NGF: 29.52 ± 0.91% VEGF: 58.56 ± 1.31%	NGF: 4 days VEGF: 11 days	In vitro results showed an increase in cell proliferation; In vivo results of NGC with NGF + VEGF were not similar to those of the autograft but better than the results of NGC without growth factors or NGC with a single growth factor;	[81]
NGF	PCL nanofibers loaded with NGF through heparin	Max 3 ng	28 days	Gradient favors neurite growth with preferential direction; NGC with gradient concentration of NGF promoted regeneration in terms of axonal regeneration and remyelination compared to NGC without gradient; NGC with gradient promoted motor functional recovery compared to NGC without gradient and NGC without NGF;	[78]
Melatonin (MLT)	PCL nanofibers loaded with melatonin and magnetic nanoparticles (S1: NGC with a single layer of nanofibers; S2: NGC with multi-layers)	S1: 54% S2: 80%	21 days	S1 and S2 stimulated the expression of S100 (a marker of SCs) and NF200, a marker of neurofilament, to a greater extent than PCL NGC without MLT and magnetic nanoparticles; PCL nanofibers reduced the expression of vimentin that indicates the activity of fibroblasts; S1 and S2, with MLT and Magnetic nanoparticles, stimulated shift from macrophage M1 to macrophage M2 and reduced inflammatory process; S1 and, especially, S2 promoted nerve regeneration in terms of morphology and electrophysiology;	[86]

## Data Availability

Not applicable.

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
