# Peer review of "Recent Advances in Polymeric Drug Delivery Systems for Peripheral Nerve Regeneration"

_pharmaceutics, 2023, doi:10.3390/pharmaceutics15020640_

Round 1

Reviewer 1 Report

The manuscript pharmaceutics-2194735 “Recent advances on Polymeric Drug Delivery Systems for Peripheral Nerve Regeneration” by Marta Bianchini et al. reviews recent advances in the study of biodegradable polymeric drug delivery systems for nerve tissue regeneration and also focuses on their potential to improve the effectiveness of clinical therapy for severe nerve damage. The manuscript is logical and well written. The paper will definitely be of interest to the readers of Pharmaceutics.

Questions and comments:

  1. A figure illustrating the structure of nerve tissue should be added to Section 1.1.

  2. Line 263 - Dexamethasone is one of the most commonly used corticosteroid drugs in tissue damage and inflammation, as well as in tissue and organ transplantation. A brief introduction to the mechanism of action of dexamethasone and the possibilities of its application is important. Perhaps this reference will be helpful: doi.org/10.3390/biomedicines9040341  

  3. Polymers of different nature, including natural and synthetic polymers, are used to produce polymeric drug delivery systems. Do natural polymers have advantages for the development of DDS for the treatment of nerve damage?

  4. Section 2.3.2 provides detailed information about the different types of polymeric drug delivery systems. Nevertheless, a brief conclusion to this section is necessary. The described DDS should be compared with each other. Do electrospinning matrices have advantages compared to polymeric micro- and nanoparticles?

Author Response

Please find our response in the uploaded file

Reviewer 2 Report

This review article is a significant effort by the authors to the research area in nerve regeneration. However, I have some suggestions to improve the quality of the work.

- abbreviations must be provided with the words where they appear first. e.g. Drug delivery system in Abstract.

- It would be better if you could add illustrations with processes like autograft, allograft, NGC, etc.

- kindly add an appendix section for the abbreviations as there are many in this article.

- section on polymeric microfluidic devices for nerve regeneration therapy would be good.

- Kindly confine your references to a certain time period alike the previous decade or five years. It will make the data in the review article concentrated and more meaningful for the readers.

Author Response

(The authors gave the same response as above.)

Reviewer 3 Report

The manuscript “Recent advances on Polymeric Drug Delivery Systems for Peripheral Nerve Regeneration” seems interesting and informative for researchers working on the nerve regeneration area. The overall manuscript looks good and is well-designed. The authors have discussed the useful contents and different strategies that are used for peripheral nerve regeneration. Additionally, they have also discussed the current challenges associated with them and their future perspectives. Thus, I recommend it for publication.

Author Response

We would like to thank the Reviewer#3 for this good feedback on our work.